# Perinatal Acetaminophen Exposure and Childhood Attention-Deficit/Hyperactivity Disorder (ADHD): Exploring the Role of Umbilical Cord Plasma Metabolites in Oxidative Stress Pathways

**DOI:** 10.3390/brainsci11101302

**Published:** 2021-09-30

**Authors:** Neha S. Anand, Ramkripa Raghavan, Guoying Wang, Xiumei Hong, Romuladus E. Azuine, Colleen Pearson, Barry Zuckerman, Hehuang Xie, Xiaobin Wang

**Affiliations:** 1Department of Pediatrics, Johns Hopkins University School of Medicine, Baltimore, MD 21205, USA; nanand7@jhmi.edu; 2Center on Early Life Origins of Disease, Department of Population, Family and Reproductive Health, Johns Hopkins University Bloomberg School of Public Health, Baltimore, MD 21205, USA; ramkripa@gmail.com (R.R.); gwang24@jhu.edu (G.W.); xhong3@jhu.edu (X.H.); 3Office of the Director, National Institutes of Health, Bethesda, MD 20892, USA; romuladus.azuine@nih.gov; 4Department of Pediatrics, Boston University School of Medicine and Boston Medical Center, Boston, MA 02118, USA; colleen.pearson@bmc.org (C.P.); barry.zuckerman@bmc.org (B.Z.); 5Department of Biomedical Sciences & Pathobiology, Fralin Life Sciences Institute at Virginia Technology, Blacksburg, VA 24061, USA; davidxie@vt.edu

**Keywords:** ADHD, cord blood, acetaminophen, glutathione, oxidative stress, neurodevelopment

## Abstract

Oxidative stress mechanisms may explain associations between perinatal acetaminophen exposure and childhood attention-deficit hyperactivity disorder (ADHD). We investigated whether the changes in umbilical cord plasma amino acids needed to synthesize the antioxidant glutathione and in the oxidative stress biomarker 8-hydroxy-deoxyguanosine may explain the association between cord plasma acetaminophen and ADHD in the Boston Birth Cohort (BBC). Mother–child dyads were followed at the Boston Medical Center between 1998 and 2018. Cord plasma analytes were measured from archived samples collected at birth. Physician diagnoses of childhood ADHD were obtained from medical records. The final sample consisted of 568 participants (child mean age [SD]: 9.3 [3.5] years, 315 (52.8%) male, 248 (43.7%) ADHD, 320 (56.3%) neurotypical development). Cord unmetabolized acetaminophen was positively correlated with methionine (R = 0.33, *p* < 0.001), serine (R = 0.30, *p* < 0.001), glycine (R = 0.34, *p* < 0.001), and glutamate (R = 0.16, *p* < 0.001). Children with cord acetaminophen levels >50th percentile appeared to have higher risk of ADHD for each increase in cord 8-hydroxy-deoxyguanosine level. Adjusting for covariates, increasing cord methionine, glycine, serine, and 8-hydroxy-deoxyguanosine were associated with significantly higher odds for childhood ADHD. Cord methionine statistically mediated 22.1% (natural indirect effect logOR = 0.167, SE = 0.071, *p* = 0.019) and glycine mediated 22.0% (natural indirect effect logOR = 0.166, SE = 0.078, *p* = 0.032) of the association between cord acetaminophen >50th percentile with ADHD. Our findings provide some clues, but additional investigation into oxidative stress pathways and the association of acetaminophen exposure and childhood ADHD is warranted.

## 1. Introduction

Acetaminophen is the most-commonly used medication during pregnancy for pain and fever [1,2]. Two large European cohort studies, the Norwegian Mother and Child Cohort and the Danish Birth Cohort study, were the first to link maternal report of acetaminophen use during pregnancy with adverse offspring neurodevelopmental outcomes, including attention-deficit hyperactivity disorder (ADHD) [3,4]. Subsequently, several studies in Europe, Asia, and the United States have associated perinatal acetaminophen exposure with increased risk of childhood neurodevelopmental disorders [5,6,7,8,9,10,11,12,13,14,15,16,17,18]. Recent meta-analyses of observational studies have also found significant pooled risks for childhood ADHD and hyperactivity symptoms associated with perinatal acetaminophen exposure [19,20,21,22].

Although several studies have documented associations between perinatal acetaminophen exposure and the risk of childhood ADHD, possible mechanisms to explain acetaminophen exposure and childhood neurodevelopmental outcomes have not been exhaustively investigated [23]. Gervin et al. found differences in DNA methylation in cord blood samples between children with ADHD versus controls, which were associated with prenatal exposure to acetaminophen for more than 20 days. These differences were found in genes linked to ADHD, neural development, neurotransmission, and pathways involving oxidative stress and the olfactory sensory system. Baker et al. identified altered frontoparietal–sensorimotor brain cortex connectivity as a mediator of the association between acetaminophen exposure in meconium samples and childhood measures of hyperactivity. Animal studies have suggested that acetaminophen may impact fetal development through neuro- and endocrine disruption and through pathways of oxidative stress and inflammation [1,24,25,26]. However, to date, there have been no studies in humans of biomarkers of metabolic pathways hypothesized to explain the association.

Elucidating the potential mechanism that may explain adverse childhood outcomes associated with acetaminophen exposure, such as ADHD, is of clinical and public health significance [1,2,23,27]. In this study, we leverage unique data from a 20-year longitudinal Boston Birth Cohort (BBC) to investigate the mechanisms that potentially explain the link between acetaminophen and developmental disorders, focusing on oxidative stress pathways. Recent analysis of the BBC, a prospective U.S. birth cohort, found a robust positive association between umbilical cord biomarkers of acetaminophen and childhood diagnosis of ADHD, even with multiple sensitivity analyses [6]. By using umbilical cord data as biomarkers, that study overcame limitations of previous studies that measured prenatal acetaminophen exposure by maternal self-report of acetaminophen use [6].

In the same cord samples from this study of the BBC, several analytes were measured as part of a metabolomic study. We aimed to analyze relationships of cord plasma analytes in these samples that may signal oxidative stress with cord plasma acetaminophen and childhood ADHD. Glutathione, the antioxidant that detoxifies the acetaminophen metabolite N-acetyl-p-benzoquinone imine (NAPQI) [28], was not directly available in the BBC dataset. Thus, we sought to analyze whether four cord plasma amino acids that are necessary for glutathione synthesis [29] (methionine, serine, glycine, and glutamate (Figure 1A)) were associated with cord acetaminophen and risk of childhood ADHD. We hypothesized that cord plasma methionine, serine, glycine, and glutamate would be depleted for children with higher cord plasma acetaminophen exposure due to greater demand and need for glutathione to detoxify acetaminophen. We further hypothesized that the depletion of cord amino acids may explain the association between acetaminophen exposure and childhood risk of ADHD, as these amino acids have roles in other pathways that may be affected, such as epigenetic mechanisms and neurotransmission. Secondly, we aimed to analyze whether cord plasma 8-hydroxy-deoyguanosine, a known biomarker of oxidative damage to DNA, was associated with acetaminophen exposure and ADHD risk [30,31]. 8-Hydroxy-deoxyguanosine is a product of cellular oxidative stress, which can occur when NAPQI causes mitochondrial damage that subsequently leads to the release of reactive oxygen species (Figure 1B) [32]. We hypothesized that for children with higher cord acetaminophen levels, higher 8-hydroxy-deoxyguanosine levels would be associated with ADHD risk.

## 2. Materials and Methods

### 2.1. Study Design

Data came from the Boston Birth Cohort (BBC), which is a prospective birth cohort study of 3165 mother–child pairs followed at the Boston Medical Center (BMC) from 1998 to 2018. Mother–child pairs were enrolled within 24 to 72 h of the child’s birth after written informed consent by the mother. Informed consent was obtained from all subjects involved in the study. Exclusion criteria included multiple gestation pregnancies, in vitro fertilization pregnancies, maternal trauma-induced deliveries, and infants born with major birth defects or chromosomal abnormalities. The BBC is a preterm-enriched cohort. For every preterm (born <37 weeks of gestation) or low birthweight infant (born <2500 g), about two term and normal weight infants were enrolled [33]. Children that continued primary pediatric or specialty care at BMC beginning at 6 months of age were invited for post-natal follow-up until the age of 21. The consent of children, written or verbal, was also obtained depending on their age. The initial and post-natal follow-up protocols were approved by the Institutional Review Boards of the Johns Hopkins Bloomberg School of Public Health and the BMC. All data for analyses were de-identified and accessed only by authorized investigators. The full details of the BBC have been previously published [34,35].

### 2.2. Independent Variable: Childhood ADHD

Childhood neurodevelopmental diagnoses were categorized based on clinician documentation of primary and secondary diagnoses in electronic medical records up to June 2018. The ADHD-only category included children with only ADHD-related codes (ICD-9 codes 314.0–314.9; ICD-10 codes F90.0–F90.9) and excluded children with autism spectrum disorder (ASD) diagnoses (ICD-9 codes 299.0-299.91; ICD-10 codes F84.0–F84.9) or other mental, behavioral, and neurodevelopmental disorders (ICD-9 codes 290–319; ICD-10 codes F01–F99). The neurotypical development category included children without ADHD, ASD, or other developmental disability diagnoses.

### 2.3. Dependent Variables: Umbilical Cord and Maternal Plasma Analytes

Umbilical cord blood samples were collected at birth, and maternal non-fasting plasma samples were obtained at the time of enrollment (within 24–72 h after delivery). Both maternal and cord blood samples were processed and fractionated into cells and plasma shortly after collection by the field team at the BMC. Quantitative profiling of analytes, including unmetabolized acetaminophen, methionine, serine, glycine, glutamate, and 8-hydroxy-deoxyguanosine from cord and maternal plasma samples was performed at the Harvard-MIT Broad Institute Metabolite Profiling Laboratory using liquid chromatography–tandem mass spectrometry. Due to the skewed distribution of most cord analytes, inverse normal transformation was used to render the distribution approximately normal. Details on laboratory methods, quality control, and data processing have been previously described [6,36].

### 2.4. Maternal and Child Covariates

We included covariates drawing upon previously published analyses of the BBC [6]. These included maternal age at delivery (in years); parity (nulliparous vs. multiparous); maternal race/ethnicity (Black, White, Hispanic, or Other); maternal education level (below college degree vs. above college degree); maternal body mass index (BMI); stress during pregnancy (mild, average, severe); smoking during pregnancy (never, quit, or continuous); alcohol use before or during pregnancy; marital status (not married vs. married); child sex, delivery type (cesarean vs. vaginal); preterm birth (<37 weeks); low birthweight (<2500 g); stress during pregnancy (mild, average, or severe); and maternal fever during pregnancy (yes vs. no).

### 2.5. Statistical Analyses

Our statistical analyses followed five steps. First, unmetabolized cord acetaminophen was cut at 0.0301, the 50th percentile, for analyses based on its graphical association with ADHD risk (Appendix A). Then, maternal and child characteristics for children with acetaminophen levels ≤50th percentile and vs. >50th percentile were compared with analysis of variance tests (ANOVA) for continuous characteristics and Pearson’s chi-squared test for categorical characteristics. Simple imputation was used for missing sociodemographic characteristics. The median was imputed for continuous variables, and the most frequent value was imputed for categorical variables.

Second, we calculated Pearson correlations between selected cord analytes, and unmetabolized acetaminophen as well as 8-hydroxy-deoxyguanosine were calculated. Third, we plotted the associations between childhood ADHD risk and cord unmetabolized acetaminophen stratified by above and below the 50th percentile of 8-hydroxy-deoxyguanosine. Similarly, the associations between childhood ADHD risk and cord 8-hydroxy-deoxyguanosine stratified by above and below the 50th percentile of unmetabolized acetaminophen were plotted for the total sample as well as for males and females separately. In Appendix A, we compared the odds of childhood ADHD of the following four groups for the total sample, males, and females: cord acetaminophen ≤50th percentile and 8-hydroxy-deoxyguanosine ≤50th percentile (reference), cord acetaminophen ≤50th percentile and 8-hydroxy-deoxyguanosine >50th percentile, cord acetaminophen >50th percentile and 8-hydroxy-deoxyguanosine ≤50th percentile, and cord acetaminophen >50th percentile and 8-hydroxy-deoxyguanosine >50th percentile.

Fourth, we used unadjusted and adjusted logistic regressions to examine the associations between cord acetaminophen and selected cord analytes with childhood diagnosis of ADHD with neurotypical development as the reference group. Given the positive association between unmetabolized acetaminophen and many of the selected analytes, we aimed to assess whether adjusting for analytes in regressions may attenuate the odds of ADHD for cord acetaminophen levels >50th percentile, which would signal possible statistical mediation. In Appendix A, we conducted the same regressions for maternal plasma analytes in addition to determining the Pearson correlations for maternal and selected cord plasma analytes.

Lastly, we used the VanderWeele–Vansteelandt approach to estimate the natural indirect effect of selected cord analytes for the association between cord acetaminophen and ADHD with the ‘medflex’ package in R [37]. Sensitivity analyses for the mediation analyses were performed for the data that excluded preterm births, maternal alcohol use, maternal smoking during pregnancy, and maternal fever. For all analyses, children who were siblings with another participant in the BBC were excluded to maintain independence of subjects in analyses.

## 3. Results

### 3.1. Maternal and Child Characteristics

The source population of 3165 mother–child dyads included 433 (13.7%) participants with a childhood diagnosis of ADHD only and 1444 (45.6%) with neurotypical development. Of these, 965 participants had cord plasma analyte data when excluding siblings, of which 248 (25.7%) had childhood ADHD and 320 (33.2%) had neurotypical development. Table 1 compares maternal and child characteristics for the total analytic sample of 568 dyads between those with cord unmetabolized acetaminophen levels ≤50th percentile vs. >50th percentile. Children with cord acetaminophen >50th percentile were significantly more likely to have childhood ADHD diagnosis, mothers who smoked during pregnancy, and low birthweight. Further, cord methionine, serine, glycine, and glutamate levels were higher for those with cord acetaminophen levels >50th percentile.

### 3.2. Correlations of Cord Plasma Methionine, Serine, Glycine, and Glutamate with Cord Plasma Unmetabolized Acetaminophen and 8-Hydroxy-Deoxyguanosine

Figure 2A–D display the correlations between cord unmetabolized acetaminophen and the selected cord analytes. Cord acetaminophen was positively correlated with methionine (R = 0.33, *p* < 0.001), serine (R = 0.30, *p* < 0.001), glycine (R = 0.34, *p* < 0.001), and glutamate (R = 0.16, *p* < 0.001). Figure 3A–D displays positive correlations between cord 8-hydroxy-deoxyguanosine and cord methionine (R = 0.33, *p* < 0.001), serine (R = 0.33, *p* < 0.001), glycine (R = 0.25, *p* < 0.001), and glutamate (R = 0.21, *p* < 0.001).

### 3.3. Correlations of Cord Plasma Methionine, Serine, Glycine, and Glutamate with Cord Plasma Unmetabolized Acetaminophen and 8-Hydroxy-Deoxyguanosine

In Figure 4A, children with cord 8-hydroxy-deoxyguanosine levels >50th percentile compared to below appear to have higher risk of ADHD for each increase in acetaminophen level. Further, children with cord acetaminophen levels >50th percentile compared to ≤50th percentile appear to have a higher risk of ADHD for each increase in 8-hydroxy-deoxyguanosine level (Figure 4B). This finding is consistent for both males (Figure 4C) and females (Figure 4D), although females have an overall lower risk of ADHD. Appendix A demonstrates that children with cord acetaminophen >50th percentile had significantly higher odds of ADHD when cord 8-hydroxy-deoxyguanosine levels were ≤50th percentile (OR: 2.21, 95% CI (1.37, 3.58), *p* = 0.001) and even higher odds when 8-hydroxy-deoxyguanosine was >50th percentile (OR: 2.38, 95% CI (1.49, 3.82), *p* < 0.001) compared to children with both cord acetaminophen and 8-hydroxy-deoxyguansoine levels ≤50th percentiles for the total sample. These findings were consistent for males but not females, although the sample size was reduced when stratifying.

### 3.4. Logistic Regressions of Odds of ADHD and Cord Plasma Analytes

Table 2 displays logistic regressions results of childhood ADHD diagnosis with the selected cord analytes, unadjusted and adjusted for covariates. Cord unmetabolized acetaminophen >50th percentile was associated with higher odds of ADHD diagnosis (aOR: 2.10, 95% CI (1.43, 3.11), *p* < 0.001). In adjusted regressions, increasing levels of cord methionine (aOR: 1.43, 95% CI (1.17, 1.77), *p* = 0.001), serine (aOR: 1.38, 95% CI (1.14, 1.68), *p* = 0.008), glycine (aOR: 1.38, 95% CI (1.14, 1.68), *p* = 0.001), and 8-hydroxy-deoxyguanosine (aOR: 1.24, 95% CI (1.01, 1.52), *p* = 0.039) were associated with higher odds of ADHD as well but not cord glutamate.

When including both cord acetaminophen and cord methionine or glycine in the regression model, the odds ratio for diagnosis of any ADHD for children with cord plasma acetaminophen >50th percentile vs. ≤50th percentile decreased by 14–15% (cord acetaminophen ≥50th percentile only: aOR: 2.10, 95% CI (1.43, 3.11), *p* < 0.001; with methionine: aOR: 1.79, 95% CI (1.19, 2.71), *p* = 0.005; with glycine: aOR: 1.80, 95% CI (1.19, 2.73), *p* = 0.005). The increased odds of ADHD for each unit increase of cord methionine or cord glycine also remained significant when included in the model with cord acetaminophen (methionine: aOR: 1.30, 95% CI (1.05, 1.62), *p* = 0.018; glycine: aOR: 1.26, 95% CI (1.02, 1.54), *p* = 0.030). Cord serine did not have a significant odds ratio when included in the regression with cord acetaminophen >50th percentile, and including cord glutamate or 8-hydroxy-deoxyguanosine did not substantially attenuate the odds ratio for cord acetaminophen. Appendix A displays the adjusted logistic regression results stratified by males and females. For males, when including cord acetaminophen ≥50th percentile in the regression, cord methionine, serine, and glycine were no longer significantly associated with higher odds of ADHD. However, for females, these cord analytes remained significant, and cord acetaminophen ≥50th percentile was not significantly associated with ADHD. Of note, the sample size of children with ADHD was considerably reduced for females (196 neurotypical, 57 ADHD).

### 3.5. Cord Plasma Methionine and Glycine as Partial Mediators for the Association between Cord Plasma Unmetabolized Acetaminophen and Childhood ADHD

Given the positive correlation between cord acetaminophen and cord methionine and glycine, significantly higher odds of ADHD with each unit of cord methionine or glycine, and the attenuation of the odds of ADHD for cord acetaminophen levels >50th percentile when adjusting for cord methionine or glycine in the total sample, mediation analysis was pursued. Using the VanderWeele–Vansteelandt approach, increasing cord methionine was estimated to mediate 22.1% of the association between cord acetaminophen >50th percentile on subsequent childhood ADHD (natural indirect effect log odds ratio: 0.167, SE: 0.071, *p* = 0.019) and increasing cord glycine mediated 22.0% of the association (natural indirect effect logs odds ratio: 0.166, SE: 0.078, *p* = 0.032), (Appendix A). Modeling cord acetaminophen as a continuous variable in the mediation yielded similar results for both increasing cord methionine and cord glycine.

In sensitivity analyses (Appendix A), cord methionine significantly or borderline significantly partially mediated the association between cord acetaminophen >50th percentile and childhood ADHD diagnosis when excluding preterm births (23.9%, *p* = 0.049), maternal alcohol use (18.8%, *p* = 0.056), maternal smoking (29.0%, *p* = 0.013%), or maternal fever (30.3%, *p* = 0.007). Cord glycine significantly mediated the association when excluding maternal smoking (30.1%, *p* = 0.025) and maternal fever (29.8%, *p* = 0.010) but not when excluding preterm births (19.6%, *p* = 0.149) or maternal alcohol use (18.8%, *p* = 0.071).

### 3.6. Correlations of Cord and Maternal Plasma Methionine and Glycine and Associations of Maternal Plasma Methionine and Glycine with Childhood ADHD

Lastly, maternal plasma methionine and glycine levels were compared to cord methionine and glycine levels respectively for 453 mother–child dyads with available maternal analyte data. While maternal methionine and cord methionine were significantly correlated for children with neurotypical development (R = 0.156, *p* = 0.008), they were not correlated for children with diagnoses of ADHD only (R = −0.074, *p* = 0.349). Further maternal methionine was not significantly associated with increased odds for ADHD diagnosis (aOR: 1.02, 95% CI (0.80, 1.29), *p* = 0.894), (Appendix A). Maternal glycine was correlated with cord glycine for both neurotypical (R = 0.282, *p* < 0.001) and ADHD only groups (R = 0.275, *p* < 0.001). However, maternal glycine was also not significantly associated with increased odds for ADHD diagnosis (aOR: 0.92, 95% CI (0.72, 1.17), *p* = 0.490), (Appendix A).

## 4. Discussion

### 4.1. Main Findings and Interpretation

In this subset of the prospective Boston Birth Cohort study, we hypothesized that cord plasma amino acids involved in the synthesis of the antioxidant glutathione (Figure 1A) would be decreased for children with higher cord plasma acetaminophen levels and may explain the association between acetaminophen exposure and childhood risk of ADHD. We further hypothesized that 8-hydroxy-deoxyguanosine, a biomarker of oxidative stress (Figure 1B), would be associated with ADHD risk for children with higher cord acetaminophen levels. We found that increased levels of umbilical cord plasma methionine, serine, glycine, and glutamate, which are precursors of glutathione, were correlated with cord unmetabolized acetaminophen and the oxidative stress biomarker 8-hydroxy-deoxyguanosine. For children with cord acetaminophen >50th percentile compared to ≤50th percentile, increasing 8-hydroxy-deoxyguanosine appeared to be associated with greater risk of childhood ADHD. In regressions adjusting for covariates, increasing levels of cord methionine, serine, glycine, and 8-hydroxy-deoxyguanosine, but not glutamate, were found to be associated with higher odds of childhood ADHD compared to neurotypical development. Further, both increasing cord methionine and glycine partially statistically mediated the association between cord acetaminophen levels >50th percentile and childhood ADHD diagnosis. These mediation findings were significant or borderline significant when analyzing subgroups that excluded known risk factors for ADHD such as preterm birth, maternal smoking, and maternal alcohol use [38,39].

These findings are in opposition to our original hypothesis that lower levels of amino acids involved in the synthesis of glutathione may explain the association between cord acetaminophen and ADHD risk. However, elevated levels of methionine, serine, glycine, and glutathione were significantly correlated with the oxidative stress biomarker 8-hydroxy-deoxyguanosine. This may suggest that potential disruption of the homeostasis of glutathione synthesis by acetaminophen is associated with increased oxidative stress. In a study of rats, subtoxic and toxic doses of acetaminophen led to a significant increase in 8-hydroxy-deoxyguanosine and a decrease in glutathione [32]. Indeed, the significant association of higher levels of 8-hydroxy-deoxyguanosine with higher odds of ADHD in this study is consistent with our original hypothesis of the oxidative stress biomarker being associated with childhood ADHD risk. Though oxidative stress is a hypothesized mechanism for the development of ADHD, few studies have looked at the association of 8-hydroxy-deoxyguanosine with childhood ADHD. One case-control study found higher levels of urinary 8-hydroxy-deoxyguanosine in children 6 to 12 years old with ADHD compared to those without ADHD [40]. However, another study has reported lower levels of 8-hydroxy-deoxyguanosine in children with ADHD [41]. In this sample, it appears that although higher acetaminophen exposure may be increasing ADHD risk through an independent mechanism, the presence of 8-hydroxy-deoxyguanosine, which could signal the baseline level of oxidative stress, may modify this risk.

Both cord methionine and cord glycine partially statistically mediated the association between acetaminophen and childhood ADHD. Unlike serine, glycine, and glutamate, methionine is an essential amino acid derived from dietary sources. In addition to being a precursor through glutathione through the transsulfuration pathway, methionine itself can also act as an antioxidant for reactive species [28,42]. Due to its role in antioxidation, methionine has been studied as a treatment for acetaminophen poisoning [43,44]. Methionine is also a precursor to the molecule S-adenosyl-methionine (SAM-e), which is an important methyl donor involved in epigenetic processes and the synthesis of monoamine neurotransmitters such as dopamine, serotonin, and norepinephrine [45]. SAM-e has been evaluated as a treatment of neuropsychiatric disorders, including ADHD, as well as in studies for acetaminophen poisoning [45,46,47,48,49,50]. While human studies are lacking, animal studies have found that excess methionine may affect fetal development, including neural development, and disturb pools of SAM-e, potentially affecting methylation of DNA and proteins [51]. Animal studies have also found that the restriction of methionine is associated with the stimulation of glutathione production and reduction of oxidative stress [28]. These studies suggest that alteration of fetal methionine levels may be involved in processes hypothesized to be implicated in neurodevelopmental disorders.

Increased levels of cord methionine could occur due to the higher maternal transfer of methionine across the placenta or due to the blockage of methionine in its downstream metabolic pathways. In this study, maternal dietary information was not available. However, maternal methionine and cord methionine levels were not found to be significantly correlated, and maternal methionine was not associated with increased risks for ADHD. This may point to increased cord methionine levels likely due to decreased fetal catabolism of methionine. It is possible that fetal metabolism of acetaminophen may affect the catabolism of methionine. Although the literature is scarce on this topic, one animal study found that the administration of acetaminophen led to a decreased expression of methionine adenosyltransferase (MAT), which is the enzyme that converts methionine to SAM-e in the transmethylation pathway, as well as depressed levels of SAM-e [52].

More research is needed to elucidate how acetaminophen may interfere with methionine metabolic pathways, particularly the transmethylation and transsulfuration pathways. A study by Thomas et al. (2008) on neonatal methionine metabolism found that after feeding, neonates had high rates of transmethylation, which may reflect high demand for cellular processes requiring methylation. Premature neonates, who have a greater risk for ADHD, had high rates of transsulfuration, which could be related to greater demands for glutathione [38,53]. Perinatal acetaminophen exposure may be disrupting these pathways early in life, potentially interfering with methylation processes and increasing the demand for glutathione, especially in premature infants.

Glycine, in addition to being involved in the synthesis of glutathione, is a known neurotransmitter that leads to excitatory neurotransmission as a co-agonist of the NMDA receptor in the brain and can have inhibitory effects at the spinal cord [54]. In rodents, high brain glycine levels have been associated with inhibition of dopamine release, which is thought to be through the co-stimulation of the NMDA receptor, and altered dopamine neurotransmission has been suggested to contribute to ADHD [55,56]. A study of children and college students found that higher glycine dietary intake was associated with ADHD diagnosis. However, the literature has also suggested that glycine supplementation may overcome NMDA dysregulation, which is thought to be involved in ADHD from animal and genetic studies, and treatments targeting the glycine site of the NMDA receptor have been proposed [57].

Given its role in glutathione synthesis, glycine has also been studied to overcome acetaminophen toxicity. In cell cultures of mice hepatocytes, glycine helped to prevent acetaminophen-induced necrotic cell killing [58]. In another animal study on rodents that received acetaminophen, a prodrug containing glycine prevented the depletion of glutathione and had both gastroprotective and hepatoprotective effects [59]. In the context of our study, higher cord glycine levels associated with cord acetaminophen may reflect a snapshot of when glycine was in higher demand to synthesize glutathione to overcome oxidative stress. Higher plasma glycine levels may have had an impact on neurotransmission perinatally that may contribute to ADHD development, which would explain the positive statistical mediation finding. Although cord glycine and maternal glycine were positively correlated for both ADHD and neurotypical groups, suggesting that increased glycine may be coming from the mother, maternal glycine was not associated with greater odds for ADHD. This may signal that maternal dietary glycine is not an influential factor for explaining the association between acetaminophen and ADHD, though again maternal dietary data were not available for this study.

It is important to note that a positive statistical mediation finding does not necessarily suggest a causal relationship between perinatal acetaminophen exposure and higher methionine or glycine levels. Elevated cord methionine and glycine levels may be a marker of altered catabolism due to pathways not affected by acetaminophen that are also associated with ADHD. Furthermore, this study detected partial mediation, suggesting that the association between acetaminophen exposure and ADHD mainly occurs through other direct or indirect mechanisms.

### 4.2. Strengths, Limitations, and Future Directions

To our knowledge, this is the first study that explores potential metabolic mechanisms to explain the association between perinatal acetaminophen exposure and childhood ADHD in a human sample. Major strengths include the prospective birth cohort study design and examination of multiple cord analytes involved in oxidative stress. This study offers new insight into the biological plausibility of acetaminophen toxicity on fetal neurodevelopment.

However, our study has several limitations. First, the cord plasma measurements of analytes collected at birth may reflect only a snapshot of fetal metabolism, and it is difficult to draw temporal conclusions from the correlation of the analytes. Furthermore, the relative intensities of these analytes were measured, not absolute values. Due to high rates of no detection, correlations of other cord metabolites of acetaminophen previously studied in the BBC with cord methionine could not be adequately discerned. The dataset also does not have sufficient information to correlate cord acetaminophen intensities to maternal dosage and frequency of acetaminophen use.

Second, in the main analyses presented, cord acetaminophen was modeled as a dichotomous variable with a cutoff at the 50th percentile of the sample. This was done to maximize sample size while also reflecting the graphical associations between cord acetaminophen and neurodevelopmental diagnoses. However, this may have resulted in a loss of granularity in the analyses. Third, despite adjusting for several potential cofounders, unmeasured and residual confounding may still impact the results of our analyses. For example, potential genetic confounding factors for risk of ADHD were not investigated. Acetaminophen use during late pregnancy has been associated with maternal polygenic risk scores for ADHD, which may confound the association between acetaminophen exposure and development of childhood ADHD [60]. Lastly, since this study consisted of a predominantly urban, low-income, and minority population and a preterm-enriched cohort, caution is needed to extrapolate these results to other populations.

Given the observational study design, this study should be interpreted as hypothesis generating and not causal. Our findings have implications for understanding the potential mechanism of acetaminophen toxicity on neurodevelopment and for potential shifts in guidelines for acetaminophen use during pregnancy. Driven by the growing number of research studies associating in utero acetaminophen exposure with adverse childhood outcomes, including neurodevelopmental outcomes, multiple researchers and doctors in a 2021 consensus statement called for the reassessment of the safety of acetaminophen use during pregnancy by U.S. and European regulators [61]. Further studies are needed to identify the potential pathways between perinatal acetaminophen exposure and fetal neurodevelopment as well as the long-term consequences of perinatal acetaminophen exposure. Future analyses could aim to directly analyze cord plasma SAM-e, dopamine, glutathione, or NAPQI levels, which were unavailable for this report, as well as other established biomarkers of oxidative stress as potential mediators of the association. Correlations of biological measurements of fetal acetaminophen levels to maternal usage of the drug are also lacking in the literature. Lastly, in this sample, we noted that children with higher cord acetaminophen levels were more likely to have low birthweight, which has been studied as a risk factor for ADHD [62,63,64]. Future studies could investigate low birthweight as a modifier or mediator of the perinatal acetaminophen exposure and childhood ADHD relationship.

## 5. Conclusions

In this study of the prospective Boston Birth Cohort, cord plasma amino acids involved in the synthesis of antioxidant glutathione (methionine, serine, and glycine) and the oxidative stress biomarker 8-hydroxy-deoxyguanosine were associated with increased odds of childhood ADHD. Furthermore, cord plasma methionine and glycine were statistical partial mediators of the association between higher levels of cord acetaminophen and childhood ADHD. These results suggest that oxidative stress mechanisms should be further explored to understand the link between perinatal acetaminophen exposure and risk of childhood ADHD.

## Figures and Tables

**Figure 1 brainsci-11-01302-f001:**
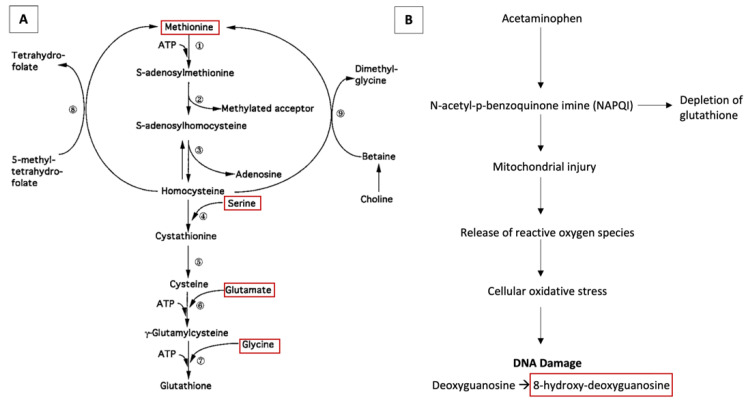
Pathways for synthesis of antioxidant glutathione (**A**) ^a^ and acetaminophen toxicity leading to oxidative stress (**B**) ^b,c^. ^a^ Figure adapted with permission from Lu, S. C. (1999) [29]. Regulation of hepatic glutathione synthesis: current concepts and controversies. *FASEB J, 13*(10), 1169–1183. ^b^ Figure based on Scheme 1 from C. L. Powell et al. [32], “Phenotypic anchoring of acetaminophen-induced oxidative stress with gene expression profiles in rat liver,” (in eng), Toxicol Sci, vol. 93, no. 1, pp. 213–22, Sep 2006, doi:10.1093/toxsci/kfl030. ^c^ Selected analytes for study boxed in red.

**Figure 2 brainsci-11-01302-f002:**
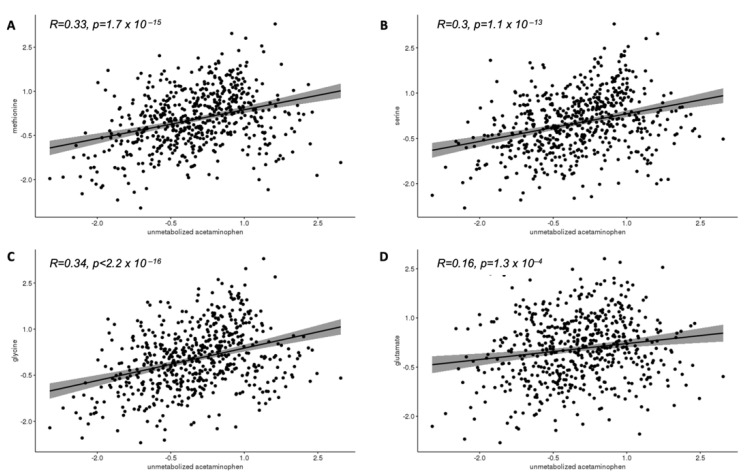
Associations of cord plasma unmetabolized acetaminophen ^a^ with cord plasma methionine (**A**), serine (**B**), glycine (**C**), and glutamate (**D**) ^a^. ^a^ Inverse normal transformed intensities.

**Figure 3 brainsci-11-01302-f003:**
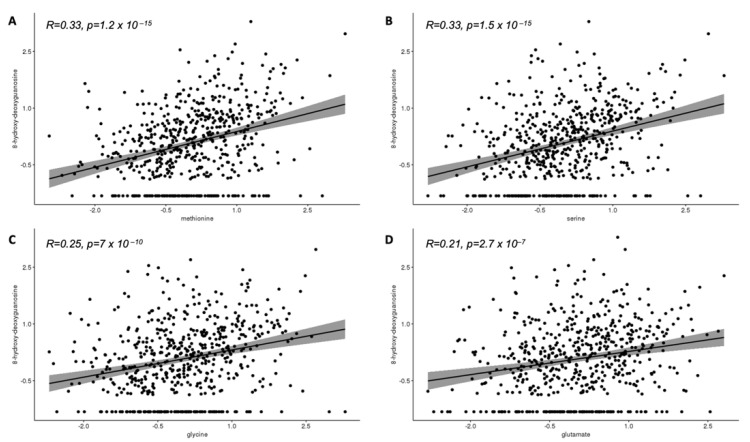
Association of cord plasma methionine (**A**), serine (**B**), glycine (**C**), and glutamate (**D**) with cord plasma 8-hydroxy-deoxyguanosine ^a^. ^a^ Inverse normal transformed intensities.

**Figure 4 brainsci-11-01302-f004:**
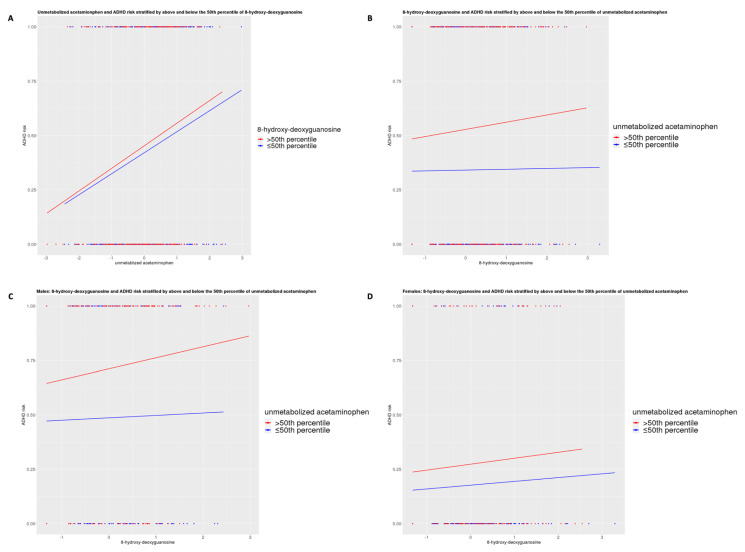
Inter-relationships of cord plasma unmetabolized acetaminophen ^a^, 8-hydroxy-deoxyguanosine ^a^, and childhood ADHD ^b^. (**A**): Cord unmetabolized acetaminophen and ADHD risk stratified by cord 8-hydroxy-deoxyguanosine above and below 50th percentile. (**B**): Cord 8-hydroxy-deoxyguanosine and ADHD risk stratified by cord unmetabolized acetaminophen above and below 50th percentile. (**C**): Males: cord 8-hydroxy-deoxyguanosine and ADHD risk stratified by cord unmetabolized acetaminophen above and below 50th percentile. (**D**): Females: cord 8-hydroxy-deoxyguanosine and ADHD risk stratified by cord unmetabolized acetaminophen above and below 50th percentile. ^a^ Inverse normal transformed intensities. ^b^ Reference group = neurotypical development.

**Table 1 brainsci-11-01302-t001:** Maternal and child characteristics by cord plasma acetaminophen ≤50th percentile vs. >50th percentile ^a^.

Characteristics	Total	Cord Acetaminophen ≤50th Percentile	Cord Acetaminophen >50th Percentile	*p*-Value ^b^	% Missing ^c^
*n*	568	284	284		
Childhood ADHD diagnosis, *n* (%)	248	97 (34.2%)	151 (53.2%)	<0.001 *	
Maternal age (years), *mean (SD)*	28.0 (6.6)	28.0 (6.4)	28.0 (6.8)	0.973	
Nulliparous, *n* (%)	243	112 (39.4%)	131 (46.1%)	0.107	
Maternal Race or ethnicity,* n* (%)				0.111	
Black	276	190 (66.9%)	186 (65.5%)		
White	32	12 (4.2%)	20 (7.0%)		
Hispanic	118	55 (19.4%)	63 (22.2%)		
Other	42	27 (9.5%)	15 (5.3%)		
Maternal Education, *n* (%)				0.855	0.53%
Below College Degree	396	199 (70.1%)	197 (69.4%)		
College Degree or Above	172	85 (29.9%)	97 (30.6%)		
Maternal BMI (kg/m^2^), *mean (SD)*	26.4 (6.1)	26.0 (6.2)	26.7 (5.9)	0.170	5.46%
Maternal smoking,*n* (%)				0.005 *	1.23%
Never	484	255 (89.8%)	229 (80.6%)		
Quit	33	14 (4.9%)	19 (6.7%)		
Continuous	51	15 (5.3%)	36 (12.7%)		
Maternal alcohol use before or during pregnancy, *n* (%)				0.107	4.05%
No	535	272 (85.8%)	263 (92.6%)		
Yes	33	12 (4.2%)	21 (7.4%)		
Marital Status, *n* (%)				0.181	0.88%
Not Married	381	183 (64.4%)	198 (69.7%)		
Married	187	101 (35.6%)	86 (30.3%)		
Child age in years by last visit, *mean (SD)*	9.3 (3.5)	9.6 (3.5)	9.1 (3.4)	0.129	
Child Sex, *n* (%)				0.205	
Male	315	150 (52.8%)	165 (58.1%)		
Female	253	134 (47.2%)	119 (41.9%)		
Delivery type, *n* (%)				0.857	0.53%
Cesarean	180	91 (32.0%)	89 (31.3%)		
Vaginal	388	193 (68.0%)	195 (68.7%)		
Preterm, *n* (%)				0.051	
No	490	253 (89.1%)	237 (83.5%)		
Yes	78	31 (10.9%)	47 (16.5%)		
Low Birthweight, *n* (%)				0.002 ^c,^*	0.18%
No	482	254 (89.4%)	228 (80.3%)		
Yes	86	30 (10.6%)	56 (19.7%)		
Stress During Pregnancy, *n* (%)				0.444	0.88%
Mild	219	116 (40.8%)	103 (36.3%)		
Average	246	121 (42.6%)	125 (44.0%)		
Severe	103	47 (16.5%)	56 (19.7%)		
Maternal Fever During Pregnancy, *n* (%)				0.844	5.11%
No	541	271 (95.4%)	270 (95.1%)		
Yes	27	13 (4.6%)	14 (4.9%)		
Cord methionine ^a^ *(mean, SD)*	0.04 (0.99)	−0.28 (0.93)	0.37 (0.94)	<0.001 *	
Cord serine ^a^ *(mean, SD)*	0.01 (0.98)	−0.31 (0.90)	0.33 (0.96)	<0.001 *	
Cord glycine ^a^ *(mean, SD)*	0.03 (1.01)	−0.33 (0.91)	0.39 (0.98)	<0.001 *	
Cord glutamate ^a^ *(mean, SD)*	0.06 (1.00)	−0.13 (1.01)	0.24 (0.95)	<0.001 *	
Cord 8-hydroxy-deoxyguanosine ^a^ *(mean, SD)*	0.08 (0.95)	0.06 (0.93)	0.10 (0.98)	0.604	

^a^ Inverse normal transformed intensities ^b^ *p*-values calculated using analysis of variance tests (continuous variables) or Pearson chi-sq tests (categorical variables) ^c^ Missing characteristics imputed with the median for continuous variables or the most frequent value for categorical variables * *p* < 0.05.

**Table 2 brainsci-11-01302-t002:** Logistic regressions examining risk of childhood ADHD for cord plasma metabolites ^a^.

			Unadjusted Logistic			Adjusted Logistic ^d^		
		*n*	Odds Ratio	95% CI	*p*-Value	Odds Ratio	95% CI	*p*-Value
**ADHD only** ^a^		568						
**regressions for individual metabolites** ^b^								
acetaminophen >50th percentile^c^			2.23	(1.59, 3.13)	<0.001 *	2.10	(1.43, 3.11)	<0.001 *
methionine			1.34	(1.13, 1.60)	0.001 *	1.43	(1.17, 1.77)	0.001 *
glycine			1.33	(1.12, 1.57)	0.001 *	1.38	(1.14, 1.68)	0.001 *
serine			1.26	(1.07, 1.50)	0.008 *	1.31	(1.07, 1.61)	0.008 *
glutamate			1.23	(1.04, 1.45)	0.018 *	1.21	(0.99, 1.47)	0.058
8-hydroxy-deoxyguanosine			1.09	(0.92, 1.30)	0.328	1.24	(1.01, 1.52)	0.039 *
**regression with both acetaminophen and methionine** ^b^								
acetaminophen >50th percentile ^c^			1.98	(1.39, 2.84)	<0.001 *	1.79	(1.19, 2.71)	0.005 *
methionine			1.20	(1.00, 1.45)	0.050	1.30	(1.05, 1.62)	0.018 *
**regression with both acetaminophen and serine** ^b^								
acetaminophen >50th percentile ^c^			2.07	(1.45, 2.97)	<0.001 *	1.89	(1.26, 2.87)	0.002 *
serine			1.12	(0.93, 1.35)	0.217	1.18	(0.95, 1.46)	0.128
**regression with both acetaminophen and glycine** ^b^								
acetaminophen >50th percentile ^c^			1.99	(1.39, 2.85)	<0.001 *	1.80	(1.19, 2.73)	0.005 *
glycine			1.18	(0.99, 1.42)	0.071	1.26	(1.02, 1.54)	0.030 *
**regression with both acetaminophen and glutamate** ^b^								
acetaminophen >50th percentile ^c^			2.12	(1.50, 3.00)	<0.001 *	2.01	(1.35, 2.99)	0.001 *
serine			1.15	(0.96, 1.37)	0.126	1.13	(0.93, 1.39)	0.219
**regression with both acetaminophen and 8-hydroxy-deoxyguanosine** ^b^								
acetaminophen >50th percentile ^c^			2.22	(1.58, 3.13)	<0.001 *	2.08	(1.41, 3.09)	<0.001 *
8-hydroxy-deoxyguanosine			1.08	(0.91, 1.29)	0.380	1.23	(1.00, 1.51)	0.051 *

^a^ Reference group = neurotypical development (N = 320); ADHD only (N = 248): children with only a neurodevelopmental diagnosis of attention-deficit hyperactivity disorder ^b^ Inverse normal transformed intensities ^c^ Reference group = cord acetaminophen level ≤ 50th percentile ^d^ Adjusted model covariates: maternal age at delivery in years, parity (nulliparous vs. multiparous), maternal race/ethnicity (Black, White, Hispanic, or Other), maternal education level (below college degree vs. above college degree), maternal body mass index (BMI), stress during pregnancy (mild, average, severe), maternal fever during pregnancy (yes, no), smoking during pregnancy (never, quit, or continuous), alcohol use before or during pregnancy, marital status (not married vs. married), child sex, delivery type (cesarean vs. vaginal), preterm birth (<37 weeks), and low birthweight (<2500 g) * *p* < 0.05.

## Data Availability

The data presented in this study are available on request from the corresponding author. The data are not publicly available due to participant privacy.

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
