# Peer review of "Perinatal Acetaminophen Exposure and Childhood Attention-Deficit/Hyperactivity Disorder (ADHD): Exploring the Role of Umbilical Cord Plasma Metabolites in Oxidative Stress Pathways"

_brainsci, 2021, doi:10.3390/brainsci11101302_

Round 1

Reviewer 1 Report

This topic is very interesting and paper is well written, please just look at these points:

  1. Lines 82-94 and Figure 1 would probably look better in the discussion section.
  2. Lines 214-216: It seems that low birthweight (P 0.002) impact of ADHD. How do authors explain this result?
  3. Lines 336-350. Please improve references. Look at these 3 very important refs: Association of Maternal Neurodevelopmental Risk Alleles With Early-Life Exposures. JAMA Psychiatry. 2019 Aug 1;76(8):834-842. doi: 10.1001/jamapsychiatry.2019.0774.    ------   Clinical Risk and Overall Survival in Patients with Diabetes Mellitus, Hyperglycemia and Glioblastoma Multiforme. A Review of the Current Literature. Int J Environ Res Public Health. 2020 Nov 17;17(22):8501. doi: 10.3390/ijerph17228501.  ------ Prenatal and postnatal exposure to acetaminophen in relation to autism spectrum and attention-deficit and hyperactivity symptoms in childhood: Meta-analysis in six European population-based cohorts. Eur J Epidemiol. 2021 May 28.
  4. Lines 342-346: "o a significant increase in 8-hydroxy-deoxyguanosine.... this study is consistent with our original hypothesis of the oxidative stress biomarker being associated with childhood ADHD risk". Please improve this concept.

Overall a good paper. Minor revision.

Author Response

This topic is very interesting and paper is well written, please just look at these points:

  1. Lines 82-94 and Figure 1 would probably look better in the discussion section.

    Thank you for this suggestion. As this section and figure provides background for the justification of investigating the selected metabolites for this study, we have decided to keep it in the Introduction section. However, we have reiterated our hypotheses and referred to Figure 1 at the beginning of the discussion section (Lines: 319-325).

  2. Lines 214-216: It seems that low birthweight (P 0.002) impact of ADHD. How do authors explain this result?

    Low birthweight and preterm birth have been studied as risk factors for ADHD, which is why we have controlled for these factors in our multivariable models.

    Table 1 shows that children with cord acetaminophen >50th percentile were more likely to have low birthweight. While this is not the focus of this current manuscript, a recent birth cohort study (Arneja J, Hung RJ, Seeto RA, Knight JA, Hewko SL, Bocking A, Lye SJ, Brooks JD. Association between maternal acetaminophen use and adverse birth outcomes in a pregnancy and birth cohort. Pediatr Res. 2020 Jun;87(7):1263-1269. doi: 10.1038/s41390-019-0726-8. Epub 2019 Dec 18. PMID: 31852009.) found an association between acetaminophen use during pre-pregnancy and increased risk of low birthweight, but other studies have found no association between acetaminophen use during pregnancy and low birthweight. Further, a mechanism for how acetaminophen exposure may contribute to low birthweight has not been well delineated.

    It would be interesting for future studies to explore if low birthweight mediates or modifies acetaminophen-ADHD association. We have added this observation and need for future research to the limitation section (Lines 472-476)

  3. Lines 336-350. Please improve references. Look at these 3 very important refs: Association of Maternal Neurodevelopmental Risk Alleles With Early-Life Exposures. JAMA Psychiatry. 2019 Aug 1;76(8):834-842. doi: 10.1001/jamapsychiatry.2019.0774.    ------   Clinical Risk and Overall Survival in Patients with Diabetes Mellitus, Hyperglycemia and Glioblastoma Multiforme. A Review of the Current Literature. Int J Environ Res Public Health. 2020 Nov 17;17(22):8501. doi: 10.3390/ijerph17228501.  ------ Prenatal and postnatal exposure to acetaminophen in relation to autism spectrum and attention-deficit and hyperactivity symptoms in childhood: Meta-analysis in six European population-based cohorts. Eur J Epidemiol. 2021 May 28.

    Thank you for these additional important references. We have added these in the following places:
  • Association of Maternal Neurodevelopmental Risk Alleles With Early-Life Exposures. JAMA Psychiatry. 2019 Aug 1;76(8):834-842. doi: 10.1001/jamapsychiatry.2019.0774.

    We cited this study when discussing the limitation of not controlling for confounding of genetic factors in the Discussion section (Lines 452-455)

  • Prenatal and postnatal exposure to acetaminophen in relation to autism spectrum and attention-deficit and hyperactivity symptoms in childhood: Meta-analysis in six European population-based cohorts. Eur J Epidemiol. 2021 May 28.

    We cited this study in the introduction in the line that discusses meta-analyses on the association between acetaminophen exposure and ADHD (Line 48)

The other reference suggested (Clinical Risk and Overall Survival in Patients with Diabetes Mellitus, Hyperglycemia and Glioblastoma Multiforme. A Review of the Current Literature. Int J Environ Res Public Health. 2020 Nov 17;17(22):8501. doi: 10.3390/ijerph17228501.) did not seem relevant to the topic of the paper, so we did not include it.

We also added a reference on the recent Nature Endocrine Review consensus statement calling for precautionary action for acetaminophen use during pregnancy in the Discussion section (Lines 461-465): Bauer, A.Z., Swan, S.H., Kriebel, D. et al. Paracetamol use during pregnancy — a call for precautionary action. Nat Rev Endocrinol (2021). https://doi.org/10.1038/s41574-021-00553-7

4. Lines 342-346: "o a significant increase in 8-hydroxy-deoxyguanosine.... this study is consistent with our original hypothesis of the oxidative stress biomarker being associated with childhood ADHD risk". Please improve this concept.

We have elaborated on this point, citing two studies that have evaluated the association between 8-hydryoxy-deoxyguanosine and childhood ADHD given the hypothesis that oxidative stress may contribute to the pathophysiology of ADHD (Lines 347-352).

Reviewer 2 Report

The article Perinatal acetaminophen exposure and childhood attention-
deficit/hyperactivity disorder (ADHD): exploring the role of 
umbilical cord plasma metabolites in oxidative stress pathways comprises a very interesting and significant observations on ADHD aetiology and pathogenesis. The aim of the study, materials and methods are presented appropriately, the background clearly explains the investigated relationship between perinatal acetamoniphen exposure and oxidative stress and ADHD. The conclusions adequately represent the results. 

Author Response

The article Perinatal acetaminophen exposure and childhood attention-
deficit/hyperactivity disorder (ADHD): exploring the role of 
umbilical cord plasma metabolites in oxidative stress pathways comprises a very interesting and significant observations on ADHD aetiology and pathogenesis. The aim of the study, materials and methods are presented appropriately, the background clearly explains the investigated relationship between perinatal acetamoniphen exposure and oxidative stress and ADHD. The conclusions adequately represent the results. 

Thank you for your review of our study and comments on the aims, methods, results, and discussion.